# All regular $4 \times 4$ solutions of the Yang–Baxter equation

Luke Corcoran[⋆] and Marius de Leeuw[†]

School of Mathematics & Hamilton Mathematics Institute,
Trinity College Dublin, Ireland

⋆ lcorcoran@maths.tcd.ie , † mdeleeuw@maths.tcd.ie

## Abstract

We complete the classification of 4 × 4 regular solutions of the Yang–Baxter equation. Apart from previously known models, we find four new models of non-difference form. All the new models give rise to Hamiltonians and transfer matrices that have a non-trivial Jordan block structure. One model corresponds to a non-diagonalisable integrable deformation of the XXX spin chain.

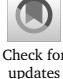

# 1 Introduction

The Yang–Baxter equation

$$R_{12}(u,v)R_{13}(u,w)R_{23}(v,w) = R_{23}(v,w)R_{13}(u,w)R_{12}(u,v) \tag{1}$$

is central to the study of quantum integrable models. This equation was first studied by Yang in the context of a one-dimensional scattering problem [1], and by Baxter in the context of the eight-vertex model [2]. Over the years (1) has appeared in many different settings, and its applications range widely from correlation functions in quantum field theory and the AdS/CFT correspondence [3,4] to statistical physics and condensed matter systems such as the Hubbard model [5].

(1) is an equation in $V_1 \otimes V_2 \otimes V_3$, where each $V_i \simeq \mathbb{C}^n$, and the indices on the $R$-matrices $R_{ij}(u,v) \in \text{End}(V_i \otimes V_j)$ indicate on which spaces they act non-trivially. An integrable spin-chain Hamiltonian $\mathbb{Q}_2(u)$ can be constructed from the $R$-matrix, and (1) directly implies the existence of a tower of charges $\mathbb{Q}_r(u)$, $r = 1,2,3,\ldots$, which pairwise commute; this is the cornerstone of integrability. The integrability of a model often allows for the use of powerful techniques to compute its spectrum exactly, for example the algebraic Bethe ansatz [6].

The prototypical example of such a quantum integrable model for $n = 2$ is the Heisenberg XXX spin-$\frac{1}{2}$ chain [7]. Its $R$-matrix admits both trigonometric and elliptic deformations to the XXZ and XYZ models respectively [8]. These $R$-matrices are of difference form $R(u-v)$, and the corresponding conserved charges $\mathbb{Q}_r$ do not depend on the spectral parameter $u$. In the most general case these charges do depend on $u$, and this parameter can be viewed as an inhomogeneity on the underlying spin chain. In this case the corresponding $R$ matrix depends independently on the spectral parameters $R(u,v)$. A particularly interesting class of solutions to (1) are so-called regular solutions, which satisfy $R(u,u) = P$. This condition ensures that the first conserved charge $\mathbb{Q}_1$ is simply the generator of translations along the spin chain, and is natural in the context of periodic chains.

Given its ubiquity in physics, it is an interesting and challenging problem to solve the Yang–Baxter equation (1). New solutions could correspond to new integrable condensed matter systems, or novel integrable deformations of previously known models. A priori solving (1) directly is a difficult task, given that it is a set of cubic functional equations for the entries of $R$. However, several approaches have emerged. For example, one could require the $R$-matrix to have some algebra symmetry [9–11]. In this case the number of free functions in the $R$-matrix is reduced and it may be possible to classify all solutions. However, this approach does not help in cases where the model has an unknown symmetry, or possibly no symmetry at all. A different algebraic approach to solving the equation, known as Baxterisation, first emerged in the realm of knot theory [17, 18]. This approach is intimately tied to the theory of quantum groups and and consists of constructing solutions of (1) as representations of certain algebras, for example Hecke algebras and Temperly-Lieb algebras. This approach has led to numerous new insights and solutions of the Yang–Baxter equation [19–25].

Here we discuss a bottom-up approach to finding solutions of (1), which has been developed in a series of papers by one of the authors [26–29]. The main idea is to use the Hamiltonian density $\mathcal{H}(u)$, which is simply related to the $R$-matrix, as the starting point. From this density the total Hamiltonian $\mathbb{Q}_2(u)$ can be constructed, as well as the higher charge $\mathbb{Q}_3(u)$ using the so-called boost operator [30–32]. Then one can solve the Reshetikhin condition $[\mathbb{Q}_2(u), \mathbb{Q}_3(u)] = 0$ [33] to identify integrable Hamiltonians.[1] Finally, the $R$-matrix corresponding to an integrable Hamiltonian can then be constructed by solving the Sutherland equations [34], which follow from (1) and constitute a set of first order differential equations

---

[1]Although this condition is only necessary for integrability it is believed to be sufficient, and all evidence so far confims this.

for the entries of $R(u, v)$, supplemented by appropriate boundary conditions. This approach has been used to compute many new regular $R$-matrices for the cases of local Hilbert space dimension $n = 2, 3, 4$. A full classification for any $n$ has so far been missing, mainly due to the complexity of the equations resulting from the Reshetikhin condition. The most progress was made for the case $n = 2$, where all Hermitian solutions were fully classified [29]. The main result of the present paper is the completion of the classification for $n = 2$. We remark that other approaches for solving the Yang–Baxter equation using the Sutherland equations have appeared recently [35–37], as well as other approaches starting from the Hamiltonian [38–40]. A similar classification has also been completed for constant $R$ matrices [41].

Since all the Hermitian solutions have been classified, the new models we find all correspond to non-Hermitian Hamiltonians. Although such models are often regarded as unphysical, this is far from the case. For one, they can be used to model dissipative or chiral processes, see [42] and references within for an excellent review. There has also been recent interest in non-Hermitian models in the context of logarithmic conformal field theory [43, 44]. A concrete example of such a theory is the fishnet model, which arises in a strong-twist limit of $\mathcal{N} = 4$ super Yang–Mills [45]. There, owing to a chiral four-point vertex, the dilatation operator of this theory becomes non-Hermitian and in particular non-diagonalisable in certain operator sectors [46,47]. A particular 3-scalar sector of interest leads to the integrable eclectic model [48], the spectrum of which has been studied using various methods [49–52].

The paper is organised as follows. In section 2 we fix our conventions and give the main equations relevant for non-difference form $R$-matrices. In section 3 we discuss the classification of $R$-matrices and outline our approach for identifying the remaining $4 \times 4$ regular solutions to the Yang–Baxter equation. In section 4 we present our new solutions, and discuss some of their properties. We also present the remaining solutions which have been classified previously. Finally, in section 5 we conclude and give some outlook.

## 2 Yang–Baxter equation and integrability

We are studying solutions of the Yang–Baxter equation

$$R_{12}(u, v)R_{13}(u, w)R_{23}(v, w) = R_{23}(v, w)R_{13}(u, w)R_{12}(u, v), \tag{2}$$

where $R : \mathbb{C}^n \otimes \mathbb{C}^n \to \mathbb{C}^n \otimes \mathbb{C}^n$ is the $R$-matrix. A solution $R(u, v)$ of (2) is called *regular* if

$$R(u, u) = P, \tag{3}$$

where $P$ is the permutation operator on $\mathbb{C}^n \otimes \mathbb{C}^n$, which acts as[2]

$$P\, x \otimes y = y \otimes x, \qquad x, y \in \mathbb{C}^n. \tag{4}$$

Each regular solution to (2) generates an integrable nearest-neighbour Hamiltonian $\mathbb{Q}_2(u) : V \to V$, where $V := (\mathbb{C}^n)^{\otimes L}$, and $L$ is the length of the corresponding spin chain. Explicitly, this operator can be realised as a sum over Hamiltonian densities

$$\mathbb{Q}_2(u) := \sum_{i=1}^{L} \mathcal{H}_{i,i+1}(u), \tag{5}$$

---

[2]If $v \in \mathbb{C}^n \otimes \mathbb{C}^n$ is not a pure state, then this definition is extended by linearity.

where the density $\mathcal{H}_{i,i+1}(u) : \mathbb{C}^n \otimes \mathbb{C}^n \to \mathbb{C}^n \otimes \mathbb{C}^n$ is related to the $R$-matrix via[3]

$$\mathcal{H}(u) := P\frac{d}{du}R(u,v)\Big|_{v\to u}. \tag{6}$$

We implement periodic boundary conditions by identifying $\mathcal{H}_{L,L+1} \equiv \mathcal{H}_{L,1}$.

From the $R$-matrix one can form a transfer matrix

$$t(u,v) := \mathrm{tr}_a(R_{aL}(v,u)R_{a,L-1}(v,u)\cdots R_{a1}(v,u)), \tag{7}$$

and define range $r$ local operators $\mathbb{Q}_r(u) : V \to V$ from $t(u,v)$ via logarithmic derivatives

$$\mathbb{Q}_{r+1}(u) = \frac{d^r}{dv^r}\log t(u,v)\Big|_{v\to u}. \tag{8}$$

The Yang–Baxter equation (2) implies that the two-parameter transfer matrices (7) commute at different values of $v$

$$[t(u,v), t(u,v')] = 0, \tag{9}$$

which further implies that the operators $\mathbb{Q}_r(u)$ mutually commute

$$[\mathbb{Q}_r(u), \mathbb{Q}_s(u)] = 0. \tag{10}$$

Since all the operators $\mathbb{Q}_r(u)$ mutually commute, we will also refer to them as charges. The existence of an infinite number of conserved charges is one of the key features of an integrable system, and it is the Yang–Baxter equation (2) which ensures this.

The higher charges $\mathbb{Q}_3(u), \mathbb{Q}_4(u), \ldots$ can also be constructed interatively from lower charges vanish

$$\mathbb{Q}_{r+1}(u) = [\mathcal{B}[\mathbb{Q}_2(u)], \mathbb{Q}_r(u)], \qquad r = 2, 3, \ldots, \tag{11}$$

where

$$\mathcal{B}[\mathbb{Q}_2(u)] := \partial_u - \sum_{n=-\infty}^{\infty} n\mathcal{H}_{n,n+1}(u) \tag{12}$$

is the *boost operator*, introduced in [30] and first applied to non-difference form models in [31]. (12) is only well-defined on open spin chains of infinite length, however the commutator (11) is well-defined on periodic chains with a finite length. For example, if $L = 4$, then

$$\mathbb{Q}_3(u) = [\mathcal{B}[\mathbb{Q}_2(u)], \mathbb{Q}_2(u)] = \partial_u\mathbb{Q}_2(u) - \sum_{j=1}^{4}[\mathcal{H}_{j,j+1}(u), \mathcal{H}_{j+1,j+2}(u)]. \tag{13}$$

A solution $R(u,v)$ of (2) is of *difference form* if it depends only on the combination $u - v$. In this case the operators $\mathbb{Q}_r$ are independent of $u$. A solution is of *eight-vertex* type if $R$ takes the form

$$R(u,v) = \begin{pmatrix} r_{1,1}(u,v) & 0 & 0 & r_{1,4}(u,v) \\ 0 & r_{2,2}(u,v) & r_{2,3}(u,v) & 0 \\ 0 & r_{3,2}(u,v) & r_{3,3}(u,v) & 0 \\ r_{4,1}(u,v) & 0 & 0 & r_{4,4}(u,v) \end{pmatrix}, \tag{14}$$

and it is of *six-vertex* type if furthermore $r_{1,4}(u,v) = r_{4,1}(u,v) = 0$. We will call a solution $R(u,v)$ Hermitian if the corresponding Hamiltonian density $\mathcal{H}$ is Hermitian.

---

[3]It is possible to add any operator of the form $\mathcal{A}\otimes I - I \otimes \mathcal{A}$ to $\mathcal{H}$ without affecting $\mathbb{Q}_2$. We will always take $\mathcal{A} = 0$.

# 3  $R$-matrix classification

We wish to classify all regular solutions $R(u, v)$ of (2) for $n = 2$. Much progress has already been made in this direction. In particular all difference form solutions have been classified in [27], and all non-difference form solutions of six- and eight-vertex type have been classified in [29]. Furthermore, all Hermitian solutions of non-difference form have been classified, and are equivalent to known models of eight-vertex type and lower. We wish to complete the classification, and find all non-difference form $4 \times 4$ $R$-matrices. Since all Hermitian solutions are already known, any new solutions we find will necessarily correspond to non-Hermitian Hamiltonians $\mathcal{H}(u)$.

## 3.1  Identifications

Given a regular solution $R(u, v)$ to (2), there are several ways to generate more regular solutions, namely via local basis transformations, reparametrisations, normalisations, and discrete transformations, which we describe now.

**Local basis transformations.**  Given a regular solution $R(u, v)$ of (2) and a non-singular matrix $W(u) : \mathbb{C}^2 \to \mathbb{C}^2$, then

$$R^W(u, v) := [W(u) \otimes W(v)]R(u, v)[W(u) \otimes W(v)]^{-1} \tag{15}$$

is another regular solution. The corresponding Hamiltonian density is related to the Hamiltonian density $\mathcal{H}(u)$ of $R(u, v)$ via

$$\mathcal{H}^W(u) = [W(u) \otimes W(u)]\mathcal{H}(u)[W(u) \otimes W(u)]^{-1} - (\dot{W}W^{-1} \otimes I - I \otimes \dot{W}W^{-1}), \tag{16}$$

where $\dot{W} := dW/du$. Note that the transformed Hamiltonian density picks up a range-one term of the form $\mathcal{A} \otimes I - I \otimes \mathcal{A}$, which vanishes in the full Hamiltonian

$$\mathbb{Q}_2^W(u) = (W(u))^{\otimes L} \mathbb{Q}_2(u)(W(u)^{-1})^{\otimes L}. \tag{17}$$

**Reparametrisation.**  Given a regular solution $R(u, v)$ of (2), then clearly

$$R^f(u, v) := R(f(u), f(v)) \tag{18}$$

is also a solution for an arbitrary function $f : \mathbb{C} \to \mathbb{C}$. The corresponding Hamiltonian density is related to the original via

$$\mathcal{H}^f(u) = \dot{f}\,\mathcal{H}(f(u)). \tag{19}$$

**Normalisation.**  We can normalise solutions by an arbitrary function $g : \mathbb{C}^2 \to \mathbb{C}$ to get another solution

$$R^g(u, v) := g(u, v)R(u, v), \tag{20}$$

and regularity in maintained provided

$$g(u, u) = 1. \tag{21}$$

The corresponding Hamiltonian density is related to the original via

$$\mathcal{H}^g(u) = \mathcal{H}(u) + \dot{g}\,I, \tag{22}$$

where $I$ is the identity matrix, and

$$\dot{g} = \left.\frac{d}{du} g(u, v)\right|_{v \to u}. \tag{23}$$

**Discrete transformations.** Given a regular solution $R(u, v)$ to (2) with Hamiltonian density $\mathcal{H}(u)$, then $R^T(u, v), PR(v, u)P$ and $PR^T(v, u)P$ are also regular solutions, with corresponding Hamiltonian densities $P\mathcal{H}(u)^T P$, $-P\mathcal{H}(u)P$, and $-\mathcal{H}^T(u)$ respectively.

**Twists.** Given a regular $R$-matrix $R(u, v)$ and a non-singular matrix $U(u) : \mathbb{C}^n \to \mathbb{C}^n$ satisfying $[U(u) \otimes U(v), R_{12}(u, v)] = 0$, then

$$U_2(u)R_{12}(u, v)U_1(v)^{-1} \tag{24}$$

is another regular solution of (2). The corresponding transformation of the Hamiltonian is given by

$$\mathcal{H}_{12}^{\text{twist}}(u) = U_1(u)\mathcal{H}_{12}U_1(u)^{-1} + \dot{U}_1(u)U_1(u)^{-1}. \tag{25}$$

We note that there are numerous transformations on the $R$-matrix which can be given the name 'twist'. The various possibilities are discussed in more detail in [29]. Since these are all model dependent transformations, we cannot use them to refine our ansatz for integrable Hamiltonians $\mathcal{H}$, as described in section 3.3.

## 3.2 Method

Our goal is to find all regular solutions $R(u, v)$ of (2), modulo the above identifications. Our strategy follows that of [27,29]. We first identify all *potentially-integrable* Hamiltonian densities $\mathcal{H}(u)$. These are Hamiltonians such that $\mathbb{Q}_2(u) = \sum_i \mathcal{H}_{i,i+1}(u)$ commutes with the corresponding charge $\mathbb{Q}_3(u)$ constructed using the boost operator (12)

$$[\mathbb{Q}_2(u), [\mathcal{B}[\mathbb{Q}_2(u)], \mathbb{Q}_2(u)]] = 0. \tag{26}$$

In fact, all known potentially-integrable Hamiltonians $\mathcal{H}(u)$ have so far proven to be integrable, in the sense that they can be derived from a regular $R$-matrix satisfying the Yang–Baxter equation (2). Therefore we will refer to a Hamiltonian densitiy $\mathcal{H}(u)$ such that (26) is satisfied as integrable. To find the $R$-matrix corresponding to an integrable Hamiltonian density $\mathcal{H}(u)$ we use the Sutherland equations [34]

$$[R_{13}(u, v)R_{23}(u, v), \mathcal{H}_{12}(u)] = \dot{R}_{13}(u, v)R_{23}(u, v) - R_{13}(u, v)\dot{R}_{23}(u, v), \tag{27}$$

$$[R_{13}(u, v)R_{12}(u, v), \mathcal{H}_{23}(v)] = R_{13}(u, v)R'_{12}(u, v) - R'_{13}(u, v)R_{12}(u, v), \tag{28}$$

where $\dot{\ }$ and $'$ denote derivatives with respect to $u$ and $v$ respectively. (27) follows immediately from the Yang–Baxter equation (2) by taking a derivative with respect $u$ and sending $v \to u, w \to v$. Similarly, (28) follows by taking a derivative with respect to $v$ and sending $w \to v$. Given $\mathcal{H}$, these are a pair of first order differential equations for the matrix elements of $R(u, v)$, which can be solved subject to the pair of boundary conditions (3) and (6).

Since there are many equivalent $R$-matrices under the above identifications, there are also many equivalent integrable Hamiltonians. We will call integrable Hamiltonians $\mathcal{H}$ and $\tilde{\mathcal{H}}$ equivalent if they can be derived from equivalent $R$-matrices. To summarise, our approach to identify all new regular solutions of (2) is as follows:

- Parametrise a general Hamiltonian density $\mathcal{H}(u)$.

- Use identifications to simplify this ansatz, and set $\mathbb{Q}_2(u) = \sum_{i=1}^{L} \mathcal{H}_{i,i+1}(u)$. For practical purposes we can use $L = 4$.[4]

- Compute the corresponding charge $\mathbb{Q}_3(u)$ using the boost operator (13).

---

[4]For $L < 4$ cancellations can occur in $[\mathbb{Q}_2, \mathbb{Q}_3]$ which do not happen in general.

- Impose $[\mathbb{Q}_2(u), \mathbb{Q}_3(u)] = 0$, and solve these equations for the entries of $\mathcal{H}(u)$. This will give a list of integrable Hamiltonians $\mathbb{H} = \{\mathcal{H}_i(u)\}_{i=1}^N$.

- Refine the list $\mathbb{H}$, removing Hamiltonians which are equivalent to another Hamiltonian in the list, as well as Hamiltonians which correspond to an $R$-matrix previously classified in [27,29]. This will give a new list $\mathbb{H}' = \{\mathcal{H}_i(u)\}_{i=1}^M$, with $M < N$.

- For each integrable Hamiltonian $\mathcal{H}_i(u) \in \mathbb{H}'$, use the Sutherland equations (27) and (28) and the boundary conditions (3) and (6) to find the corresponding regular $R$-matrix.

- Verify that this $R$-matrix satisfies the Yang–Baxter equation (2).

### 3.3  Parametrisation of $\mathcal{H}$

We begin with a general putative regular solution $R(u, v)$ to the Yang-Baxter equation (2). Such an $R$-matrix has a Hamiltonian density (6) where all entries are non-vanishing in general. We parametrise these entries as follows

$$\mathcal{H}_{16v} = \mathcal{H}_{8v} + \tilde{\mathcal{H}}_{8v}, \tag{29}$$

where $\mathcal{H}_{8v}$ is a general eight-vertex Hamiltonian

$$\begin{aligned}
\mathcal{H}_{8v} =\; & h_1 I \otimes I + h_2(\sigma_z \otimes I - I \otimes \sigma_z) + h_3 \sigma_+ \otimes \sigma_- + h_4 \sigma_- \otimes \sigma_+ \\
& + h_5(I \otimes \sigma_z + \sigma_z \otimes I) + h_6 \sigma_z \otimes \sigma_z + h_7 \sigma_+ \otimes \sigma_+ + h_8 \sigma_- \otimes \sigma_-,
\end{aligned} \tag{30}$$

and $\tilde{\mathcal{H}}_{8v}$ accounts for the remaining entries

$$\begin{aligned}
\tilde{\mathcal{H}}_{8v} =\; & h_9(\sigma_- \otimes I + I \otimes \sigma_-) + h_{10}(\sigma_- \otimes I - I \otimes \sigma_-) + h_{11}(\sigma_+ \otimes I + I \otimes \sigma_+) \\
& + h_{12}(\sigma_+ \otimes I - I \otimes \sigma_+) + h_{13}(\sigma_- \otimes \sigma_z + \sigma_z \otimes \sigma_-) + h_{14}(\sigma_- \otimes \sigma_z - \sigma_z \otimes \sigma_-) \\
& + h_{15}(\sigma_+ \otimes \sigma_z + \sigma_z \otimes \sigma_+) + h_{16}(\sigma_+ \otimes \sigma_z - \sigma_z \otimes \sigma_+),
\end{aligned} \tag{31}$$

where we suppress the $u$ dependence of the functions $h_i$.

We remark that although $h_2, h_{10}$, and $h_{12}$ are coefficients of operators of the form $\mathcal{A} \otimes I - I \otimes \mathcal{A}$ which vanish in the full Hamiltonian $\mathbb{Q}_2$, we cannot automatically set them to zero because these functions will in general appear in the operator $\mathbb{Q}_3$ using the boost construction (13).

Since we are only interested in $R$-matrices modulo the identifications described in section 3.1, we can use these identifications to simplify our ansatz for $\mathcal{H}$. This is very important in the present situation since a priori the condition $[\mathbb{Q}_2, \mathbb{Q}_3] = 0$ is a set of first order differential equations for 16 unknown functions $h_i$. By using identifications to simplify our ansatz (29) from the beginning we can reduce the complexity of the problem significantly.

**Local basis transformations.**   First of all, we can perform a local basis transformation (15) on the $R$-matrix to modify the entries $h_7$ and $h_8$. Under such a transformation, the Hamiltonian transforms as (16). We parametrise the local change of basis as

$$W = \begin{pmatrix} a & b \\ c & d \end{pmatrix} \tag{32}$$

with $ad - bc = 1$, where we suppress all dependence on $u$. The entries $h_7$ and $h_8$ transform[5]

---

[5]The range one piece $\dot{W}W^{-1} \otimes I - I \otimes \dot{W}W^{-1}$ in (16) does not contribute to $\tilde{h}_7$ or $\tilde{h}_8$.

$$h_7 \rightarrow \tilde{h}_7 = a^4 h_7 + b^4 h_8 + 4ab^3 h_{13} - 4a^3 b\, h_{15} + a^2 b^2 (4h_6 - h_3 - h_4), \tag{33}$$

$$h_8 \rightarrow \tilde{h}_8 = c^4 h_7 + d^4 h_8 + 4cd^3 h_{13} - 4c^3 d h_{15} + c^2 d^2 (4h_6 - h_3 - h_4). \tag{34}$$

We see that for general $h_i$ we can pick $a, b, c, d$ to ensure that $\tilde{h}_7 = \tilde{h}_8 = 0$, since this amounts to the solution of a pair of quartic equations.[6] If $h_7 = h_8 = 0$ from the beginning we do not perform the basis transformation. The only situation when it is not possible to set both $\tilde{h}_7$ and $\tilde{h}_8$ to zero is when $h_{13} = h_{15} = 0$, $4h_6 = h_3 + h_4$, and either $h_7 = 0$ or $h_8 = 0$. Therefore we can restrict to two types of Hamiltonians[7]

- **Type 1**: $h_7 = h_8 = 0$.

- **Type 2**: $h_7 = h_{13} = h_{15} = 0$, $4h_6 = h_3 + h_4$, $h_8 \neq 0$.

We can also set $h_2 = 0$ with a further local basis transformation

$$W = \begin{pmatrix} a & 0 \\ 0 & a^{-1} \end{pmatrix}, \qquad a = \exp\left( \int_1^u h_2(t) dt \right). \tag{35}$$

Since $W$ is diagonal this local basis transformation does not interfere with the type 1 and type 2 constraints above.

**Shifting by the identity.** A normalisation of the $R$-matrix (20) corresponds to a shift of the Hamiltonian by the identity matrix (22). For both type 1 and type 2 Hamiltonians we can normalise the corresponding $R$-matrix by

$$g(u, v) = 1 + \int_u^v h_1(t) dt. \tag{36}$$

Comparing with (22) we see that this shifts the Hamiltonian by $-h_1 I \otimes I$, effectively setting $h_1 = 0$.

**Normalisation.** A reparametrisation of the $R$-matrix (18) corresponds to a normalisation of the Hamiltonian (19). We will use this freedom to set a single nonzero entry of $\mathcal{H}$ to 1. For type 1 Hamiltonians we don't know which specific entries will be nonzero, and so further analysis is required before this normalisation freedom can be used. For type 2 Hamiltonians we use normalisation to set $h_8 = 1$.

In summary, we have used our identifications to restrict our ansatz (29) to two different cases:

$$\mathcal{H}_{\text{type 1}} = \mathcal{H}_{16v}(h_1 = h_2 = h_7 = h_8 = 0), \tag{37}$$

$$\mathcal{H}_{\text{type 2}} = \mathcal{H}_{16v}(h_1 = h_2 = h_7 = h_{13} = h_{15} = 0,\ 4h_6 = h_3 + h_4,\ h_8 = 1). \tag{38}$$

With these ansätze the integrability condition $[\mathbb{Q}_2, \mathbb{Q}_3] = 0$ can be solved exactly.

---

[6]Due to the symmetry between the equations $(a, b) \leftrightarrow (c, d)$ the solutions of both equations are equivalent.
[7]Without loss of generality we can set $h_7 = 0$ and $h_8 \neq 0$. The case $h_8 = 0, h_7 \neq 0$ is an equivalent Hamiltonian.

### 3.4 Solving the equations

For both Hamiltonian ansätze (37) and (38) we form the total Hamiltonian $\mathbb{Q}_2(u) = \sum_{i=1}^{4} \mathcal{H}_{i,i+1}(u)$ on a periodic chain of length $L = 4$ and the corresponding charge $\mathbb{Q}_3(u)$ using the boost operator (13). The integrability condition $[\mathbb{Q}_2(u), \mathbb{Q}_3(u)] = 0$ constitutes a set of first order differential equations for the functions $h_i(u)$. We first solve the set of equations as algebraic equations for $h_i$ and $\dot{h}_i$ using the `Mathematica` command `Solve`. This is straightforward for type 2 Hamiltonians. For type 1 Hamiltonians it is necessary to consider the following 10 cases (we define $h_\pm := h_3 \pm h_4$):

- **Case 1**: $h_- = 0, h_+ \neq 0$.

- **Case 2**: $h_- \neq 0, h_+ = 0$.

- **Case 3**: $h_- = h_+ = 0, h_5 \neq 0$.

- **Case 4**: $h_- = h_+ = h_5 = 0, h_6 \neq 0$.

- **Case 5**: $h_- = h_+ = h_5 = h_6 = 0$.

- **Case 6**: $h_-, h_+ \neq 0, h_{15} = 0, h_{13} \neq 0$.

- **Case 7**: $h_-, h_+ \neq 0, h_{13} = 0, h_{15} \neq 0$.

- **Case 8**: $h_-, h_+ \neq 0, h_{13} = h_{15} = 0, h_6 \neq 0$.

- **Case 9**: $h_-, h_+ \neq 0, h_6 = h_{13} = h_{15} = 0$.

- **Case 10**: $h_-, h_+, h_{13}, h_{15} \neq 0$.

For each of these cases at least one $h_i$ is taken to be nonzero, and so we can use normalisation freedom to set one of these entries to 1. After solving the equations algebraically in $h_i$ and $\dot{h}_i$, we solve the resulting differential equations to fix the functions $h_i(u)$ and find on the order of 100 integrable Hamiltonians $\mathcal{H}(u)$. Many of these are equivalent to Hamiltonians already classified in [27, 29]. For example, any constant solution $\dot{h}_i = 0$ is equivalent to an integrable Hamiltonian classified in [27]. Furthermore, many of the non-constant solutions can be mapped to a six-/eight-vertex model under a local basis transformation (16), and thus are equivalent to an integrable Hamiltonian classified in [29]. For example, the solution

$$\mathcal{H}(u) = \begin{pmatrix} 1 & h_9(u) - \frac{(2h_9(u)+1)\dot{h}_9(u)}{4h_9(u)+1} & h_9(u) + \frac{(2h_9(u)+1)\dot{h}_9(u)}{4h_9(u)+1} & 0 \\ 1 - \frac{2\dot{h}_9(u)}{4h_9(u)+1} & 0 & 0 & h_9(u) + \frac{(2h_9(u)+1)\dot{h}_9(u)}{4h_9(u)+1} \\ 1 + \frac{2\dot{h}_9(u)}{4h_9(u)+1} & 0 & 0 & h_9(u) - \frac{(2h_9(u)+1)\dot{h}_9(u)}{4h_9(u)+1} \\ 0 & 1 + \frac{2\dot{h}_9(u)}{4h_9(u)+1} & 1 - \frac{2\dot{h}_9(u)}{4h_9(u)+1} & -1 \end{pmatrix}$$
(39)

appears to be a non-constant integrable Hamiltonian which is not of six- or eight-vertex form. However, if we take the local basis transformation

$$W(u) = \begin{pmatrix} e^{\sqrt{4h_9(u)+1}} & \frac{1}{2}e^{\sqrt{4h_9(u)+1}}\left(\sqrt{4h_9(u)+1}-1\right) \\ 1 & \frac{1}{2}\left(-\sqrt{4h_9(u)+1}-1\right) \end{pmatrix}$$
(40)

then using (16) the Hamiltonian transforms

$$\mathcal{H}(u) \to \mathcal{H}^W(u) = \begin{pmatrix} \sqrt{4h_9(u)+1} & 0 & 0 & 0 \\ 0 & 0 & 0 & 0 \\ 0 & 0 & 0 & 0 \\ 0 & 0 & 0 & -\sqrt{4h_9(u)+1} \end{pmatrix},$$
(41)

which is a trivial diagonal model. We also discard Hamiltonians which can be obtained as a specialisation of another one by fixing free functions and constants.

**Type 2 example.** As an example of the solution and identification procedure, the most non-trivial solution after solving the type 2 equations algebraically in $h_i$ and $\dot{h}_i$ is

$$\{h_-, h_{14}, h_{10}, h_9, h_5, \dot{h}_-, \dot{h}_{14}, \dot{h}_9, \dot{h}_5\} = 0,$$
$$\dot{h}_{16} = -h_{11}\left(h_+ + 4h_{16}^2\right), \qquad \dot{h}_+ = -4h_{11}h_{16}h_p. \tag{42}$$

We see that the differential equations for $\dot{h}_-, \dot{h}_{14}, \dot{h}_9, \dot{h}_5$ are trivially satisfied, leaving non-trivial differential equations for $h_{16}$ and $h_+$. The solution to these equations is

$$h_{16}(u) = \frac{-H_{11}(u)}{2(A - H_{11}(u)^2)}, \qquad h_+(u) = \frac{1}{2(A - H_{11}(u)^2)}, \tag{43}$$

where $A$ is a constant and $\dot{H}_{11} = h_{11}$.[8] The corresponding integrable type 2 Hamiltonian is

$$\mathcal{H} = \begin{pmatrix} \frac{1}{8(A-H_{11}^2)} & \frac{H_{11}}{2(A-H_{11}^2)} - h_{12} + \dot{H}_{11} & \frac{-H_{11}}{2(A-H_{11}^2)} + h_{12} + \dot{H}_{11} & 1 \\ 0 & \frac{-1}{8(A-H_{11}^2)} & \frac{1}{4(A-H_{11}^2)} & \frac{H_{11}}{2(A-H_{11}^2)} + h_{12} + \dot{H}_{11} \\ 0 & \frac{1}{4(A-4H_{11}^2)} & \frac{-1}{8(A-H_{11}^2)} & \frac{-H_{11}}{2(A-H_{11}^2)} - h_{12} + \dot{H}_{11} \\ 0 & 0 & 0 & \frac{1}{8(A-H_{11}^2)} \end{pmatrix}. \tag{44}$$

We can use identifications to bring this Hamiltonian into a nicer form. Applying the local basis transformation (16)

$$W = \begin{pmatrix} 1 & \int_1^u h_{12}(u)\,dt \\ 0 & 1 \end{pmatrix}, \tag{45}$$

sets $h_{12} = 0$. Next we make a shift by the identity matrix

$$\mathcal{H} \to \mathcal{H} - \frac{1}{8(A - H_{11}^2)} I, \tag{46}$$

and restore the normalisation we previously fixed by reparametrising $u \to \int_1^u f(t)\,dt$ and multiplying by $f(u)$, see (19). We then make the redefinitions

$$G(u) = H_{11}\left(\int_1^u f(t)\,dt\right), \qquad f(u) \to 4f(u)(A - G(u)^2), \tag{47}$$

and the resulting Hamiltonian is

$$\mathcal{H} = \begin{pmatrix} 0 & \dot{G} + 2Gf & \dot{G} - 2Gf & 4f\left(A - G^2\right) \\ 0 & -f & f & \dot{G} + 2Gf \\ 0 & f & -f & \dot{G} - 2Gf \\ 0 & 0 & 0 & 0 \end{pmatrix}, \tag{48}$$

where we suppressed the $u$ dependence in $f$ and $G$. This is the only new type 2 solution we find; the other solutions are specialisations of this Hamiltonian or can be mapped to a previously classified model. Using the Hamiltonian (48) the Sutherland equations (27) and (28) can be solved straightforwardly.

Going through all the cases, we find 4 non-constant solutions which are not possible to map to

---

[8]We can absorb an integration constant into the function $H_{11}$.

six-/eight-vertex form using our identifications. One of these is the type 2 solution (48), and 3 are solutions which arise from solving the type 1 equations for case 4 and case 5 above. The Hamiltonians with the most non-trivial (elliptic) functional dependence, occur for $h_\pm \neq 0$, i.e. cases 6–10. We find that all of these Hamiltonians can be mapped to six-/eight-vertex form. We list the new integrable Hamiltonians and their corresponding non-difference form $R$-matrices in section 4.1. For completeness, we also include the remaining $4 \times 4$ $R$-matrices which complete the classification in section 4.2 and section 4.3.

# 4 Solutions

In this section, we give the full classification of $4 \times 4$ solutions of the Yang–Baxter equation. We checked in each case explicitly that (2) is satisfied. We will present one representative of the solutions in each family. In particular, one can act on each of these models with the identifications described in section 3.1: local basis transformations, reparameterisations, normalisations, and discrete transformations. In previous papers we also used identifications using twists. We did not find any relations between the new models using these twist degrees of freedom. We have written each of the solutions in the simplest form we could find; this may have involved performing identifications after solving the equations. We denote free functions appearing in our solutions as

$$F := F(u), \qquad G := G(u), \qquad H := H(u), \tag{49}$$

i.e. we suppress $u$ dependence unless it is necessary to distinguish between $u$ and $v$. We denote their derivatives as

$$f := \dot{F}(u), \qquad g := \dot{G}(u), \qquad h := \dot{H}(u). \tag{50}$$

For brevity we will also use the shorthand

$$\delta X := X(u) - X(v). \tag{51}$$

Our models also depend on constants, which will be denoted by $A, B, C$. Finally, we choose to normalise our $R$-matrices such that the $(1,1)$ component is 1. This is possible because all our $R$-matrices are regular, and in particular the $(1,1)$ component is non-vanishing.

## 4.1 New solutions: non-difference form $R$ beyond six-/eight-vertex type

Let us first list the new models that complete the classification. In each case we checked that the corresponding $R$-matrix satisfies the Yang–Baxter equation (2).

**Model 1 – Nilpotent A.** The first $R$-matrix we find is

$$R = \begin{pmatrix} 1 & \delta G & \delta F & A\,\delta F\,\delta G \\ 0 & 0 & 1 & A\,\delta G \\ 0 & 1 & 0 & A\,\delta F \\ 0 & 0 & 0 & 1 \end{pmatrix}, \tag{52}$$

where $A$ is a constant and $F, G$ are free functions. We notice that this model is of quasi-difference form since the dependence on the spectral parameter of the entries is always of the form $X(u) - X(v)$. However, since it depends on two functions, it is not possible to use a

reparameterisation to make it depend solely on $u - v$. The corresponding Hamiltonian density is

$$\mathcal{H} = \begin{pmatrix} 0 & g & f & 0 \\ 0 & 0 & 0 & Af \\ 0 & 0 & 0 & Ag \\ 0 & 0 & 0 & 0 \end{pmatrix} \tag{53}$$

$$= \frac{g}{2}(A_+ I \otimes \sigma_+ + A_- \sigma_z \otimes \sigma_+) + \frac{f}{2}(A_+ \sigma_+ \otimes I + A_- \sigma_+ \otimes \sigma_z), \tag{54}$$

where $A_\pm := A \pm 1$.

**Model 2 – Nilpotent B.**   The next $R$-matrix is

$$R = \begin{pmatrix} 1 & \delta G & \delta F & \delta H + \frac{1}{2}\delta G^2 + \frac{1}{2}\delta F^2 + F(u)G(v) - G(u)F(v) \\ 0 & 0 & 1 & \delta F \\ 0 & 1 & 0 & \delta G \\ 0 & 0 & 0 & 1 \end{pmatrix}. \tag{55}$$

The corresponding Hamiltonian is

$$\mathcal{H} = \begin{pmatrix} 0 & g & f & h + fG - Fg \\ 0 & 0 & 0 & g \\ 0 & 0 & 0 & f \\ 0 & 0 & 0 & 0 \end{pmatrix} \tag{56}$$

$$= \frac{f+g}{2}(\sigma_+ \otimes I + I \otimes \sigma_+) + \frac{f-g}{2}(\sigma_+ \otimes \sigma_z - \sigma_z \otimes \sigma_+) + (h + fG - Fg)\sigma_+ \otimes \sigma_+. \tag{57}$$

**Model 3.**   The third $R$-matrix is

$$R = \begin{pmatrix} 1 & G(u)e^{\delta F} - G(v) & G(v)e^{\delta F} - G(u) & 2\sinh(\delta F) \\ 0 & 0 & e^{\delta F} & 0 \\ 0 & e^{\delta F} & 0 & 0 \\ 0 & 0 & 0 & 1 \end{pmatrix} \tag{58}$$

with Hamiltonian

$$\mathcal{H} = \begin{pmatrix} 0 & g + fG & -g + fG & 2f \\ 0 & f & 0 & 0 \\ 0 & 0 & f & 0 \\ 0 & 0 & 0 & 0 \end{pmatrix} \tag{59}$$

$$= \frac{f}{2}(1 - \sigma_z \otimes \sigma_z + 4\sigma_+ \otimes \sigma_+) + \frac{g + fG}{2}((1 + \sigma_z) \otimes \sigma_+) + \frac{-g + fG}{2}(\sigma_+ \otimes (1 + \sigma_z)). \tag{60}$$

**Model 4 – XXX deformation.**   The final $R$-matrix we find is

$$R = \frac{1}{1 + \delta F} \begin{pmatrix} 1 + \delta F & \delta G + 2G(u)\delta F & \delta G - 2G(v)\delta F & X(u, v) \\ 0 & \delta F & 1 & \delta G - 2G(v)\delta F \\ 0 & 1 & \delta F & \delta G + 2G(u)\delta F \\ 0 & 0 & 0 & 1 + \delta F \end{pmatrix}, \tag{61}$$

with

$$X(u, v) := \delta G^2 + A\delta F^2 + \delta F(A - 4G(u)G(v)), \tag{62}$$

where $A$ is a constant. The corresponding Hamiltonian is

$$\mathcal{H} = \begin{pmatrix} 0 & g+2fG & g-2fG & 4f(A-G^2) \\ 0 & -f & f & g+2fG \\ 0 & f & -f & g-2fG \\ 0 & 0 & 0 & 0 \end{pmatrix} \tag{63}$$

$$= \frac{f}{2}(\sigma_z \otimes \sigma_z - 1 + 2\sigma_+ \otimes \sigma_- + 2\sigma_- \otimes \sigma_+) + 4f(A-G^2)\sigma_+ \otimes \sigma_+ - 2fG(\sigma_+ \otimes \sigma_z - \sigma_z \otimes \sigma_+)$$

$$+ g(\sigma_+ \otimes I + I \otimes \sigma_+). \tag{64}$$

When $A = G = 0$ we recover the XXX model.

## 4.2 Difference form $R$ beyond six-/eight-vertex type

Many models from our previous classification in [27] are now actually given as a special case of one of the new non-difference models. Only two models remain and we will list them here.

**Model 5.** The first $R$-matrix we find is Class 3 from [27], which in our current conventions reads

$$R = \begin{pmatrix} 1 & C(e^{(2A-B)u}-1) & C(e^{(2A+B)u}-1) & 0 \\ 0 & 0 & e^{(2A+B)u} & 0 \\ 0 & e^{(2A-B)u} & 0 & 0 \\ 0 & 0 & 0 & 1 \end{pmatrix} \tag{65}$$

Since these models are of difference form, the functional dependence of the $R$-matrix is greatly simplified. In particular, we simply let the functional dependence be on $u$ for these types of models.

$$\mathcal{H} = \begin{pmatrix} 0 & (2A-B)C & (2A+B)C & 0 \\ 0 & 2A-B & 0 & 0 \\ 0 & 0 & 2A+B & 0 \\ 0 & 0 & 0 & 0 \end{pmatrix} \tag{66}$$

$$= A(1 - \sigma_z \otimes \sigma_z) + \frac{B}{2}(I \otimes \sigma_z - \sigma_z \otimes I)$$

$$+ \frac{C}{2}\left((2A+B)(\sigma_+ \otimes (1+\sigma_z)) + (2A-B)((1+\sigma_z) \otimes \sigma_+)\right). \tag{67}$$

**Model 6 – 11 vertex.** The second $R$-matrix of difference form is Class 6 from [27] and is given by

$$R = \begin{pmatrix} 1 & Au & Au & -A^2u^2(u+1) \\ 0 & \frac{u}{u+1} & \frac{1}{u+1} & -Au \\ 0 & \frac{1}{u+1} & \frac{u}{u+1} & -Au \\ 0 & 0 & 0 & 1 \end{pmatrix} \tag{68}$$

This model is the 11-vertex model [53, 54]. It can be obtained as a singular limit of the usual 8-vertex model [55].

$$\mathcal{H} = \begin{pmatrix} 0 & A & A & 0 \\ 0 & -1 & 1 & -A \\ 0 & 1 & -1 & -A \\ 0 & 0 & 0 & 0 \end{pmatrix} \tag{69}$$

$$= \frac{1}{2}(\sigma_z \otimes \sigma_z - 1 + 2\sigma_+ \otimes \sigma_- + 2\sigma_- \otimes \sigma_+) + A(\sigma_+ \otimes \sigma_z + \sigma_z \otimes \sigma_+). \tag{70}$$

Notice that this model is a deformation of the XXX spin chain, similarly to Model 4 above.

## 4.3 $R$ of six-/eight-vertex type

All models of 8-vertex type were derived in [26, 29]. For completeness we list the results here.

### 4.3.1 Difference form

First there are two models that are of difference form and they correspond to the standard XXZ and XYZ spin chains.

**6vA.** This is the usual XXZ spin chain with standard twists included

$$
R = \begin{pmatrix}
1 & 0 & 0 & 0 \\
0 & \frac{e^A \cosh B}{\sinh B \cot \delta F - 1} & \frac{\sinh B}{\sinh B \cos \delta F - \sin \delta F} & 0 \\
0 & \frac{\sinh B}{\sinh B \cos \delta F - \sin \delta F} & \frac{e^{-A} \cosh B}{\sinh B \cot \delta F - 1} & 0 \\
0 & 0 & 0 & 1
\end{pmatrix}.
\tag{71}
$$

We find the standard twisted XXZ Hamiltonian

$$
\mathcal{H} = \frac{f}{\sinh B} \begin{pmatrix}
0 & 0 & 0 & 0 \\
0 & 1 & e^{-A} \cosh B & 0 \\
0 & e^A \cosh B & 1 & 0 \\
0 & 0 & 0 & 0
\end{pmatrix}
\tag{72}
$$

$$
= \frac{f}{2 \sinh B} \left[ 1 - \sigma_z \otimes \sigma_z + 2 \cosh B \left( e^{-A} \sigma_+ \otimes \sigma_- + e^A \sigma_- \otimes \sigma_+ \right) \right].
\tag{73}
$$

**8vA.** The next model is the usual XYZ spin chain with $R$-matrix given by

$$
R = \begin{pmatrix}
1 & 0 & 0 & B \operatorname{sn}(A|B^2) \operatorname{sn}(\delta F|B^2) \\
0 & \frac{\operatorname{sn}(\delta F|B^2)}{\operatorname{sn}(\delta F + A|B^2)} & \frac{\operatorname{sn}(A|B^2)}{\operatorname{sn}(A + \delta F|B^2)} & 0 \\
0 & \frac{\operatorname{sn}(A|B^2)}{\operatorname{sn}(A + \delta F|B^2)} & \frac{\operatorname{sn}(\delta F|B^2)}{\operatorname{sn}(\delta F + A|B^2)} & 0 \\
B \operatorname{sn}(A|B^2) \operatorname{sn}(\delta F|B^2) & 0 & 0 & 1
\end{pmatrix},
\tag{74}
$$

where $\operatorname{sn}, \operatorname{cn}, \operatorname{dn}$ are the Jacobi elliptic functions with modulus $B$. The corresponding Hamiltonian reads

$$
\mathcal{H} = f \begin{pmatrix}
0 & 0 & 0 & B \operatorname{sn}(A|B^2) \\
0 & -\frac{\operatorname{cn}(A|B^2) \operatorname{dn}(A|B^2)}{\operatorname{sn}(A|B^2)} & \frac{1}{\operatorname{sn}(A|B^2)} & 0 \\
0 & \frac{1}{\operatorname{sn}(A|B^2)} & -\frac{\operatorname{cn}(A|B^2) \operatorname{dn}(A|B^2)}{\operatorname{sn}(A|B^2)} & 0 \\
B \operatorname{sn}(A|B^2) & 0 & 0 & 0
\end{pmatrix}
\tag{75}
$$

$$
= f \frac{\operatorname{cn}(A|B^2) \operatorname{dn}(A|B^2)}{2 \operatorname{sn}(A|B^2)} \left[ \sigma_z \otimes \sigma_z - 1 + \frac{1 + B \operatorname{sn}(A|B^2)^2}{\operatorname{cn}(A|B^2) \operatorname{dn}(A|B^2)} \sigma_x \otimes \sigma_x + \frac{1 - B \operatorname{sn}(A|B^2)^2}{\operatorname{cn}(A|B^2) \operatorname{dn}(A|B^2)} \sigma_y \otimes \sigma_y \right].
\tag{76}
$$

### 4.3.2 Non-difference form

There are also 6- and 8-vertex models that have an $R$-matrix that is of non-difference form. These models all satisfy the free fermion condition [56] and are related to the integrable structures that appear in the lower dimensional instances of the AdS/CFT correspondence [26, 29].

**6vB.** We have the free fermion model

$$
R = \begin{pmatrix}
1 & 0 & 0 & 0 \\
0 & \frac{e^A \delta F}{1+G(u)\delta F} & \frac{1}{1+G(u)\delta F} & 0 \\
0 & \frac{1}{1+G(u)\delta F} & e^{-A}\left[G(v) - \frac{G(u)}{1+\delta F G(u)}\right] & 0 \\
0 & 0 & 0 & \frac{1-G(v)\delta F}{1+G(u)\delta F}
\end{pmatrix}
\tag{77}
$$

with Hamiltonian

$$
\mathcal{H} = f \begin{pmatrix}
0 & 0 & 0 & 0 \\
0 & -G & G^2 - g & 0 \\
0 & 1 & -G & 0 \\
0 & 0 & 0 & -2G
\end{pmatrix}
\tag{78}
$$

$$
= f\left[\sigma^- \otimes \sigma^+ + (G^2 - g)\sigma^+ \otimes \sigma^- - \tfrac{G}{2}(\sigma^z \otimes 1 + 1 \otimes \sigma^z - 2)\right].
\tag{79}
$$

It is related to a solution of the coloured Yang–Baxter equation [57].

**8vB.** We also have a free fermion model of 8-vertex type

$$
R = \begin{pmatrix}
1 & 0 & 0 & \frac{A\,\mathrm{cn}\,\mathrm{sn}\sqrt{\sin G(u)}\sqrt{\sin G(v)}}{\mathrm{cn}\sin\frac{\Sigma G}{2} - \mathrm{dn}\,\mathrm{sn}\cos\frac{\Sigma G}{2}} \\
0 & \frac{\mathrm{cn}\sin\frac{\delta G}{2} + \mathrm{dn}\,\mathrm{sn}\cos\frac{\delta G}{2}}{\mathrm{cn}\sin\frac{\Sigma G}{2} - \mathrm{dn}\,\mathrm{sn}\cos\frac{\Sigma G}{2}} & \frac{\mathrm{dn}\sqrt{\sin G(u)}\sqrt{\sin G(v)}}{\mathrm{cn}\sin\frac{\Sigma G}{2} - \mathrm{dn}\,\mathrm{sn}\cos\frac{\Sigma G}{2}} & 0 \\
0 & \frac{\mathrm{dn}\sqrt{\sin G(u)}\sqrt{\sin G(v)}}{\mathrm{cn}\sin\frac{\Sigma G}{2} - \mathrm{dn}\,\mathrm{sn}\cos\frac{\Sigma G}{2}} & \frac{\mathrm{cn}\sin\frac{\delta G}{2} - \mathrm{dn}\,\mathrm{sn}\cos\frac{\delta G}{2}}{\mathrm{dn}\,\mathrm{sn}\cos\frac{\Sigma G}{2} - \mathrm{cn}\sin\frac{\Sigma G}{2}} & 0 \\
\frac{A\,\mathrm{cn}\,\mathrm{sn}\sqrt{\sin G(u)}\sqrt{\sin G(v)}}{\mathrm{cn}\sin\frac{\Sigma G}{2} - \mathrm{dn}\,\mathrm{sn}\cos\frac{\Sigma G}{2}} & 0 & 0 & 1 + \frac{2\mathrm{dn}\,\mathrm{sn}}{\mathrm{cn}\tan\frac{\Sigma G}{2} - \mathrm{dn}\,\mathrm{sn}}
\end{pmatrix}
\tag{80}
$$

where the Jacobi elliptic functions $\mathrm{sn},\mathrm{cn},\mathrm{dn}$ are with arguments $\delta F$ and with modulus $A$. Finally we introduced $\Sigma G = G(u) + G(v)$. The Hamiltonian for this model is

$$
\mathcal{H} = \begin{pmatrix}
0 & 0 & 0 & Af \\
0 & f\cot(G) & \left(f - \tfrac{g}{2}\right)\csc(G) & 0 \\
0 & \left(f + \tfrac{g}{2}\right)\csc(G) & f\cot(G) & 0 \\
Af & 0 & 0 & 2f\cot(G)
\end{pmatrix}
\tag{81}
$$

$$
= \frac{f}{\tan G} - \frac{f}{2\tan G}\left[\sigma_z \otimes 1 + 1 \otimes \sigma_z\right] + Af\left[\sigma_+ \otimes \sigma_+ + \sigma_- \otimes \sigma_-\right] + \frac{2f - g}{2\sin G}\sigma_+ \otimes \sigma_- + \frac{2f + g}{2\sin G}\sigma_- \otimes \sigma_+.
\tag{82}
$$

**Off-diagonal model.** Finally there is a purely off-diagonal model given by

$$
R = \begin{pmatrix}
\cosh\delta F & 0 & 0 & \sin\delta G \\
0 & -\sinh\delta F & \cos\delta G & 0 \\
0 & \cos\delta G & \sinh\delta F & 0 \\
\sin\delta G & 0 & 0 & \cosh\delta F
\end{pmatrix},
\tag{83}
$$

with corresponding Hamiltonian

$$
\mathcal{H} = \begin{pmatrix}
0 & 0 & 0 & g \\
0 & 0 & f & 0 \\
0 & -f & 0 & 0 \\
g & 0 & 0 & 0
\end{pmatrix}
\tag{84}
$$

$$
= f(\sigma^+ \otimes \sigma^- - \sigma^- \otimes \sigma^+) + g(\sigma^+ \otimes \sigma^+ + \sigma^- \otimes \sigma^-).
\tag{85}
$$

## 4.4 Properties of the new models

We remark on a few of the properties of the new models.

**Non-diagonalisable.** As expected, all of the new models we find are non-Hermitian. Models 1 and 2 are nilpotent, and all models are non-diagonalisable in general. We present the Jordan block spectra of all models on a chain of length $L = 4$, denoting the $\ell \times \ell$ Jordan block with generalised eigenvalue $\lambda$ as

$$
J_\lambda^\ell =
\begin{pmatrix}
\lambda & 1 & & & 0 \\
& \lambda & 1 & & \\
& & \lambda & \ddots & \\
& & & \ddots & 1 \\
0 & & & & \lambda
\end{pmatrix}.
\tag{86}
$$

For model 1 (53) we have

$$
\mathbb{Q}_2^{\text{model 1}}(u) \simeq J_0^5 \oplus (J_0^3)^{\oplus 3} \oplus (J_0^1)^{\oplus 2}.
\tag{87}
$$

For various choices of the parameter $A$ and free functions $f$ and $g$ the Jordan block spectra may refine. For example, if we take $A = -2 - \sqrt{3}$ then we find

$$
\mathbb{Q}_2^{\text{model 1}}(u) \simeq_{A \to -2 - \sqrt{3}} (J_0^3)^{\oplus 5} \oplus J_0^1.
\tag{88}
$$

Model 2 (56) has the same Jordan block spectrum as model 1 for generic choices of the free functions $f, g, h$:

$$
\mathbb{Q}_2^{\text{model 2}}(u) \simeq J_0^5 \oplus (J_0^3)^{\oplus 3} \oplus (J_0^1)^{\oplus 2}.
\tag{89}
$$

Models 1 and 2 are non-diagonalisable for all possible specialisations, because they are always nilpotent. For model 3 (59) we have

$$
\mathbb{Q}_2^{\text{model 3}}(u) \simeq (J_{-4f}^1)^{\oplus 2} \oplus (J_{-2f}^4)^{\oplus 3} \oplus J_0^2.
\tag{90}
$$

There are several interesting specialisations of this model. For example, if $G(u) = \exp(\pm i F(u))$ or $G(u) = \pm 1$ then the Jordan block spectrum refines

$$
\mathbb{Q}_2^{\text{model 3}}(u) \simeq_{G \to e^{\pm iF}} (J_{-4f}^1)^{\oplus 2} \oplus (J_{-2f}^1)^{\oplus 3} \oplus (J_{-2f}^3)^{\oplus 3} \oplus J_0^2,
\tag{91}
$$

$$
\mathbb{Q}_2^{\text{model 3}}(u) \simeq_{G \to \pm 1} (J_{-4f}^1)^{\oplus 2} \oplus J_{-2f}^1 \oplus J_{-2f}^2 \oplus (J_{-2f}^3)^{\oplus 3} \oplus (J_0^1)^{\oplus 2}.
\tag{92}
$$

Model 3 is also non-diagonalisable for all possible specialisations. For model 4 (63) we have

$$
\mathbb{Q}_2^{\text{model 4}}(u) \simeq J_{-6f}^1 \oplus J_{-4f}^3 \oplus J_{-2f}^1 \oplus (J_{-2f}^3)^{\oplus 2} \oplus J_0^5.
\tag{93}
$$

The Jordan block spectrum refines for $G \to 1$ and $G \to 0$

$$
\mathbb{Q}_2^{\text{model 4}}(u) \simeq_{G \to 1} J_{-6f}^1 \oplus J_{-4f}^1 \oplus J_{-4f}^2 \oplus J_{-2f}^1 \oplus (J_{-2f}^3)^{\oplus 2} \oplus J_0^2 \oplus J_0^3,
\tag{94}
$$

$$
\mathbb{Q}_2^{\text{model 4}}(u) \simeq_{G \to 0} J_{-6f}^1 \oplus J_{-4f}^1 \oplus J_{-4f}^2 \oplus (J_{-2f}^1)^{\oplus 7} \oplus J_0^2 \oplus J_0^3,
\tag{95}
$$

and if we simultaneously take $G \to 0$ and $A \to 0$ then the model is diagonalisable

$$
\mathbb{Q}_2^{\text{model 4}}(u) \simeq_{G, A \to 0} J_{-6f}^1 \oplus (J_{-4f}^1)^{\oplus 3} \oplus (J_{-2f}^1)^{\oplus 7} \oplus (J_0^1)^{\oplus 5}.
\tag{96}
$$

**Higher Charges.** When the total Hamiltonian $\mathbb{Q}_2(u)$ is non-diagonalisable, we find that the higher charge $\mathbb{Q}_3(u)$ is too. For example, for model 4 (63) we find that

$$\mathbb{Q}_3^{\text{model 4}}(u) \simeq J_{-2\dot{f}}^2 \oplus J_{-2\dot{f}}^3 \oplus J_{-2\dot{f}}^1 \oplus J_{-4\dot{f}}^3 \oplus J_{-6\dot{f}}^1 \oplus J_{-2\dot{f}-4if^2}^3 \oplus J_{-2\dot{f}+4if^2}^3. \tag{97}$$

We see that in this case all of the eigenvalues are of the form $\alpha \dot{f} + \beta f^2$. This is expected, since all the eigenvalues of $\mathbb{Q}_2$ are proportional to $f$, and $\mathbb{Q}_3$ takes the schematic form $\mathbb{Q}_3 \sim \partial_u \mathbb{Q}_2 - [\mathcal{H}, \mathcal{H}]$.

**Braiding unitarity.** All of the models we find satisfy the braiding unitarity condition

$$R_{12}(u,v)R_{21}(v,u) = \beta(u,v) I \tag{98}$$

with $\beta(u,v) = 1$ for the normalisations we have chosen. Renormalising any of our $R$-matrices $R(u,v) \to g(u,v)R(u,v)$ leads to a braiding factor $\beta(u,v) = g(u,v)g(v,u)$.

**Integrable deformations.** All of the models we find can be regarded as integrable deformations of previously studied models in the literature, in particular those classified in [27, 29]. Interestingly, model 4 (61) is a non-diagonalisable integrable deformation of the XXX model, since it can be written as

$$R(u,v) = \frac{R_{\text{XXX}}(\delta F)}{1 + \delta F} + \frac{\Delta R(u,v)}{1 + \delta F}, \tag{99}$$

with the well-known XXX $R$-matrix

$$R_{\text{XXX}}(u) = u I + P. \tag{100}$$

The non-difference integrable deformation $\Delta R(u,v)$ takes the form

$$\Delta R(u,v) = \begin{pmatrix} 0 & \delta G + 2G(u)\delta F & \delta G - 2G(v)\delta F & \delta G^2 + A\delta F^2 + \delta F(A - 4G(u)G(v)), \\ 0 & 0 & 0 & \delta G - 2G(v)\delta F \\ 0 & 0 & 0 & \delta G + 2G(u)\delta F \\ 0 & 0 & 0 & 0 \end{pmatrix}, \tag{101}$$

where again we have abbreviated $\delta F := \delta F(u,v) = F(u) - F(v)$ and $\delta G := \delta G(u,v) = G(u) - G(v)$. For $G \to 0$ and $A \to 0$ we see that $\Delta R(u,v) \to 0$. This explains the diagonalisability of the model in this limit (96).

## 5 Conclusions and outlook

In this paper we provided the first full classification of regular solutions of the Yang–Baxter equation for the case of local Hilbert space dimension $n = 2$. We found four new integrable models of non-difference form, all giving rise to non-Hermitian and furthermore non-diagonalisable Hamiltonians.

A natural next step would be to attempt to complete the classification of solutions for $n = 3$ and higher. The main technical obstruction for our current approach is the sheer number of free functions in a general Hamiltonian density, which is $n^4$. While identifications may be used to reduce this number somewhat, it is likely that the system of equations $[\mathbb{Q}_2, \mathbb{Q}_3] = 0$ is much more complex than the present case. However, it is possible that it is still solvable with appropriate casework. While there are some results for integrable Hamiltonians for $n = 3$ and higher [29, 36, 39], they all assume that the Hamiltonian takes a specific simple form.

Classifying the solutions in general would likely lead to many new integrable models, both Hermitian and non-Hermitian.

It would be interesting to study the new models we have found in greater detail. One could try to identify a symmetry algebra, which may hint at applications of the models. It would also be interesting to investigate if there is a way to compute the Jordan block spectra of these models using integrability methods. While it is possible to compute the Jordan block spectra symbolically for small $L$, it becomes very difficult for $L > 4$, especially when there are free functions/constants in the Hamiltonian. Given the rich algebraic structure hiding in the Yang–Baxter equation, it seems likely that there should be a version of the algebraic Bethe ansatz for non-diagonalisable models. Such a Bethe ansatz has been exhibited for the $U_q(\mathfrak{sl}_2)$ invariant XXZ spin chain, for the non-diagonalisable case where $q$ is a root of unity [58].

An enumeration of the Jordan block spectrum for the $n = 3$ integrable eclectic model has been provided in terms of $q$-binomial coefficients [49]. It would be interesting to check if the spectra of different non-diagonalisable integrable models follow similar combinatorics in terms of $q$-deformations.

Our results are also interesting in the context of integrable deformations, given that we identified a non-diagonalisable integrable deformation of the XXX spin chain (63). Given that the XXX spin chain corresponds to the dilatation operator of the $\mathfrak{su}(2)$ sector of $\mathcal{N} = 4$ super Yang–Mills [59], it seems plausible that our deformation corresponds to the dilatation operator of a deformed theory. Since our deformation is non-diagonalisable, it would correspond to a conformal field theory which is logarithmic, similar to the $\gamma$-deformation of $\mathcal{N} = 4$ [60].

# Acknowledgements

We thank C. Paletta and A.L. Retore for useful discussions.

**Funding information** MdL was supported by SFI, the Royal Society and the EPSRC for funding under grants UF160578, RGF\R1\181011, RGF\EA\180167, RF\ERE\210373 and 18/EP-SRC/3590. LC was supported by RF\ERE\210373.

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
