# Peer review of "All regular $4 \times 4$ solutions of the Yang-Baxter equation"

_SciPost Physics, doi:SciPost Phys. Core 7, 045 (2024)_

## Round 2 · Referee Report · Anonymous (Referee 1) · 2023-12-5

Strengths

  1. A classification that was lacking, with new R-matrices unknown up to now (as far as I know)
  2. Very clear

Weaknesses

  1. A bit repetitive w.r.t. the previous articles fo the authors (with collaborators)

Report

The authors perform the classification of regular $4\times 4$ $R$-matrices with spectral parameters, completing the study they started with collaborators in two previous papers, [9] and [11]. The present paper deals with non-Hermitian $R$-matrices of non-difference form (meaning the $R$-matrix depends separately on the two spectral parameters). Non-Hermitian Hamiltonian are used in statistical physics, to study non-equilibrium models, justifying the need for such a classification.

The paper is well-written, and provides interesting results, including new $R$-matrices. I think it deserves publication, and I have just few minor points to correct, that I list below.

Requested changes

  1. The paragraph around eqs (2.6)--(2.10) is a bit erratic: before (2.6) they start to explain the equation, but the full explanation comes at the end with eq. (2.10) and the sentence before: I would put directly the full explanation around eq (2.6), specially because (2.10) is in the middle of a discussion on conserved charges.

  2. They should cite the reference [12] around eq (2.12), to justify this relation.

  3. I don't understand the footnote 2: what means "not relevant"?

  4. I don't understand the discussion after eq (3.15): it is known that the telescopic terms come from a gauge transformation of the $R$-matrix, eqs (3.1), (3.2). Then, since they classify the $R$-matrices up to these transformations, why the telescopic terms should play a non-trivial role in $Q_3$?

  5. In section 4, they present the $R$-matrices they obtained. I suppose they checked that all their $R$-matrices obey the Yang--Baxter relation? It is needed, since they work with necessary conditions only. As I said, I suppose they did it, but it is not mentioned (or I missed that point): it would be better to mention it explicitly.

  6. Before section 4.1, they mention that they normalize the $R$-matrix in such a way that the (1,1) component is 1. They should add that it is always possible because $R$ is supposed to be regular, so that the (1,1) component cannot be 0.

  7. In section 4.1, the denomination of the different models is not very clear. For the two first models, I suppose that "nilpotent" refers to the Hamiltonian, but I don't understand the name "trigonometric": does it refer to a deformation of the XXZ model, in the same way they use "deformed XXX" for the model 4? If yes, maybe "deformed XXZ" and "deformed XXX" for models 3 and 4 would be less confusing?

  8. I think the section 4.3 is not needed, it is just a rephrasing of what they already did in their previous paper(s), and it adds only confusion w.r.t. section 4.2, which provides new informations.

  9. Instead of their section 4.3, a comparison with the classification of constant $R$-matrices [20] would be more interesting. In particular, using the matrices of [20], can they perform a Baxterisation procedure to build all their spectral parameter dependent $R$-matrices? That would give some insight on the algebraic structure which underlies the models they introduce, and I don't think it is a difficult check.

  10. In section 4.4, they deal with Jordan decomposition in the case $L=4$. Do they have something to say on the general decomposition when $L$ is generic? I understand that a proof of the decomposition for $L$ generic is beyond the scope of the article, but they may have some educated guess based on computations for different values of $L$?

  11. Finally, for three-state models, there are two classes of Hamiltonians related to elliptic curves, see https://arxiv.org/pdf/1303.4010.pdf. I was not able to see the counter part of these Hamiltonians (and R-matrices) in the case of the two-state case dealt in the present paper. Could the author comment on this point?

---

## Round 2 · Referee Report · Anonymous (Referee 2) · 2023-12-18

Weaknesses

  1. No substantial implications for physics.
  2. Not mathematically clear enough as partially explained in the report.
  3. Not sufficiently put in the research context.

Report

The authors propose some new 4-by-4 matrix solutions of the
Yang-Baxter equation. These are listed in Section 4.1 of
their manuscript together with the corresponding
`local Hamiltonians'. From the exposition it is not clear if
these models have any applications in physics nor if
the fact that the R-matrices satisfies Equation (2.1) will
be of any true use in order, say, to diagonalize the
Hamiltonians or the associated transfer matrices.

Although I can imagine that the steps described in the
manuscript produce valid results, this has not become fully
clear to me from the exposition. Eq. (2.1) is not the
standard form of the Yang-Baxter equation (see e.g. the
English version of Wikipedia). The second and the third
factor on the left hand side as well as the first and the
second factor on the right hand side usually appear in
opposite order. In the Sutherland equations (3.11), (3.12)
the arguments of the R-matrices are suppressed. In this
form I cannot see if and how they follow from (2.1). It is
also not explained why these equations together with (2.2)
and (2.5) should determine a unique solution of (2.1).
The issue of the twist (see footnote 5) is not properly
discussed, also not in the cited work [11], which does not
seem to consider the most general case either. Moreover,
there are known 4-by-4 matrix solutions of the Yang-Baxter
equation which violate (2.2). Thus, even if the paper would
properly classify all joint solutions of (2.1)
and (2.2), which I doubt, the title would not be justified.

I also find it unpleasant that the authors do not put their
work in the context of the considerable and substantial work
that has been accumulated on the subject of the Yang-Baxter
equation. In particular, C. N. Yang and R. J. Baxter are
two living scientist who found the physically most relevant
4-by-4 matrix solutions of the Yang-Baxter equation and
understood part of the meaning and of the importance
of this equation. Their works are not cited. The original work
on the boost operator and on the Sutherland equation is not cited,
no connection is made with the theory of quantum groups and their
representations, with algebro-geometric approaches to solve the Yang-
Baxter equation or to Drinfeld's set-theoretic form of the equation.

Altogether I would say that the paper is not providing
enough new physical insight, its exposition is not mathematically
clear enough and the context is not explained well enough to
justify its publication in Scipost Physics.

---

## Round 3 · Referee Report · Anonymous (Referee 1) · 2024-3-5

Report

The authors have answered to my requests, I think the paper can be published.
I have only a very minor remark, that I leave to the authors, who are free to take it into acount or not:
For consistency, in view of the way they present things, I think it would be good to add a sentence asserting that the definition of the charges Q_r in eq. (2.7) is consistent with the definition of Q_2 in eq. (2.4).
As I said, I leave the choice to the authors, I don't need to see a revised version.

---

## Round 3 · Referee Report · Anonymous (Referee 3) · 2024-3-6

Report

The authors complete the classification of "regular" 4x4 solutions of the Yang-Baxter equation that was initiated by one of the authors in previous works. They construct solutions that to the best of my knowledge are new. They have a non-trivial structure in that the R matrices do not have difference form. The corresponding Hamiltonians have a Jordan block structure.
Given the condition of regularity, it is also clear that there will be other 4x4 solutions that are not captured.

This work has been assessed by two previous referees, who came to different conclusions regarding the suitability of the paper for SciPost Physics. The work is definitely novel and warrants publication, the only question is whether it fulfils the criteria for SciPost Physics. As already pointed out by the second referee, non-Hermitian "Hamiltonians" can have applications to e.g. quantum master equations. Having said this, it is far from clear whether any of the new models has the required structure to permit the construction of a CP map. Hence it is at present unclear whether these models will have interesting applications. This is one of the key criticisms of the first referee, and I share this view. I therefore think that on balance the manuscript is more appropriate for SciPost Physics Core than for SciPost Physics.

---

## Round 3 · Author Response

We would like to thank the referees for their detailed reviews. We have taken the feedback into consideration and made substantial changes to the submitted article. We reply to the comments of the reviewers separately.

Reviewer 1: We thank the reviewer for the feedback on our paper. We address the 11 points of this reviewer:

1: We agree that the discussion around (2.6)-(2.10) was a bit erratic. We modified this part of the text: we started with a definition of the transfer matrix, and described how the charges are obtained from this. We believe it is more logically sound and flows better this way.

2: We added the appropriate citation after (2.11).

3: Twists are model dependent, so we cannot use them to refine our initial ansatz for integrable Hamiltonians. We checked that none of the new models we found admit a twist which maps it into another one of our models. We have updated the footnote to give more details.

4: The logic is we start with a solution R of the Yang-Baxter equation and define the Hamiltonian density as H= P d_u R. While terms proportional to h2, h10, h12 do not appear in the integrable model Q2, they do appear in Q3 from the boost construction and are necessary to ensure [Q2,Q3]=0. By setting these to 0 from the beginning we would miss some models. We can indeed set h2=0 using a gauge transformation (3.2). However, we cannot set the others to 0 without interfering with the other gauge transformations we use, this is discussed in section 3.3.

5: We verified the Yang-Baxter equation for all of the models. We mention this now in the paper at the start of section 4.

6: We added that the (1,1) component of R is nonzero due to regularity.

7: The name 'trigonometric' is simply because of the appearance of a trigonometric function of u in the R-matrix, it has nothing to do with the trigonometric R matrix corresponding to the XXZ model. We removed the denomination 'trigonometric' due to this potential confusion.

8: We wanted to include all regular solutions to the Yang-Baxter equation in this paper for completeness and for reference. We believe it is useful to present all of these models in a uniform notation. We have clearly marked that section 4.1 is the one with new information, and that sections 4.2 and 4.3 are for completeness.

9: Baxterisation is a way to generate difference form solutions of the YBE from constant solutions. We consider non-difference form solutions, so our situation is slightly different. For example, our model 1 is given by R = P + A, where A is a nilpotent matrix. Baxterisation is usually applied to the case where A is invertible, see for example section 4.1 of 1310.5545. While trying to identify possible TL or Hecke algebra interpretations of our results is interesting, it's beyond the scope of our current paper.

10: We don't have any guesses for the Jordan block spectra at higher lengths. Computing these spectra symbolically is very computationally expensive, and the size of our Hamiltonians grow exponentially with the length of the spin chain. We can compute the spectra up to length 6, but higher than that a cleverer approach is needed. In this section we just wanted to highlight that these matrices are not diagonalisable and give a flavour or their spectra.

11: We didn't find any new elliptic models. There are, however, models containing the Jacobi elliptic functions cn and dn, and as such can related to an elliptic curve. These are mentioned in section 4.3.

Reviewer 2: We thank the reviewer for the feedback, which we have used to improve our paper. However, we strongly reject any insinuation that our results are incomplete. We have carefully carried out our analysis and double checked our results. We stand by our claim that we have found all regular 4x4 solutions of the YBE. While the physical applications for our new models are unclear at the moment, it is likely that they will find use given the recent interest in non-diagonalisable models.

There was an unfortunate typo in the Yang-Baxter equation (2.1), as the reviewer correctly pointed out. Indeed, the actual form of the Yang-Baxter equation we solve is R12(u,v)R13(u,w)23(v,w)=R23(v,w)R13(u,w)R12(u,v). The Sutherland equations we use in our paper follow from this form of the Yang-Baxter equation. We checked that all of our R-matrices satisfy this Yang-Baxter equation. We added some details to the derivation of the Sutherland equations from the Yang-Baxter equation, without suppressing any arguments. We hope that this is more clear mathematically.

The Sutherland equations (3.13) and (3.14) in the latest draft and the commutation [Q2,Q3]=0 are necessary conditions for the YBE (2.1) to hold. We first classify all potentially integrable Hamiltonian densities H by solving [Q2,Q3]=0, where Q3 is constructed from H using the boost operator. This task is simplified by first making use of gauge transformations (basis transformations, scaling, reparametrisation) at the level of the R-matrix, allowing us to restrict the entries of H. We do not include twists in our initial gauge transformations because these are model dependent, so they cannot be used on a general solution R(u,v) to simplify our ansatz for H. The only potential danger is members of our final list of integrable Hamiltonians being related by a twist transformation, which we found not to be the case.

In response to 'It is also not explained why these equations together with (2.2) and (2.5) should determine a unique solution of (2.1).' As mentioned, the Sutherland equations are a priori only necessary conditions for the YBE to hold. For each integrable Hamiltonian H we found a unique solution to this pair of first order differential equations for the entries of R(u,v), after supplying the pair of boundary conditions (2.2) and (2.5). The Sutherland equations have not been proven to be a sufficient condition for the YBE to hold; as stated in the paper this is still a conjecture. Therefore we check afterwards that each solution to the Sutherland equations (3.13) and (3.14) is indeed a solution of the YBE.

In response to 'there are known 4-by-4 matrix solutions of the Yang-Baxter equation which violate (2.2). Thus, even if the paper would properly classify all joint solutions of (2.1) and (2.2), which I doubt, the title would not be justified'. The title of the paper is 'All regular 4x4 solutions of the Yang-Baxter equation'. Regularity is exactly equation (2.2), so our title is valid. We know that there are many interesting non-regular solutions (see for example the recent paper 2401.12710), but that is beyond the scope of this work and what is possible with the boost operator.

It is not yet clear how use the Yang--Baxter equation to diagonalise the associated transfer matrices. Indeed this an important question for future research. Into this direction, there is a recent paper (2309.10044) which classifies the Jordan block structure of an non-diagonalisable integrable model using the symmetries of the R-matrix.

We have improved the historical references in the paper. We have cited the original papers of Yang and Baxter, as well as the first instances of the boost operator and the Sutherland equations. We have mentioned more algebraic approaches to solving the YBE, including Baxterisation.

---

## Round 3 · List of Changes

1: Improved historical references, cited original works of Yang/Baxter/Sutherland/Tetel'man.

2: Improved introduction - added more references on algebraic approaches to solving YBE, e.g. Baxterisation.

3: Fixed typo in (1.1) and (2.1), gave more detail on derivation of Sutherland equations.

4: Reshuffled discussion around (2.6)-(2.10) to make the logic flow better.

5: Added appropriate citation after (2.11)

6: Improved discussion on twist, upgraded footnote to short paragraph. Twists are not relevant for our approach, we just need to check that in the end that our new models are not related by them.

7: Added explicitly that we verify the YBE for all new models, added this to the "method" section.

8: Added that we can normalise the (1,1) component of R to 1 using regularity.

9: Removed trigonometric denomination of model 3 due to potential conflation with XXZ model.

---

## Editorial Decision

published